# Plasma Membrane and Organellar Targets of STIM1 for Intracellular Calcium Handling in Health and Neurodegenerative Diseases

**DOI:** 10.3390/cells10102518

**Published:** 2021-09-23

**Authors:** Valentina Tedeschi, Daniele La Russa, Cristina Franco, Antonio Vinciguerra, Diana Amantea, Agnese Secondo

**Affiliations:** 1Department of Neuroscience, Division of Pharmacology, Reproductive and Odontostomatological Sciences, School of Medicine, University of Naples “Federico II”, 80131 Napoli, Italy; valentina.tedeschi@unina.it (V.T.); antonio.vinciguerra@unina.it (A.V.); 2Section of Preclinical and Translational Pharmacology, Department of Pharmacy, Health and Nutritional Sciences, University of Calabria, 87036 Rende, Italy; daniele.larussa@unical.it (D.L.R.); diana.amantea@unical.it (D.A.); 3Department of Science and Technology—DST, Division of Pharmacology, University of Sannio, 82100 Benevento, Italy; cristina.franco@unisannio.it

**Keywords:** Ca^2+^ homeostasis, lysosomes, endoplasmic reticulum (ER), Ca^2+^-storing organelles, amyotrophic lateral sclerosis (ALS)

## Abstract

Located at the level of the endoplasmic reticulum (ER) membrane, stromal interacting molecule 1 (STIM1) undergoes a complex conformational rearrangement after depletion of ER luminal Ca^2+^. Then, STIM1 translocates into discrete ER-plasma membrane (PM) junctions where it directly interacts with and activates plasma membrane Orai1 channels to refill ER with Ca^2+^. Furthermore, Ca^2+^ entry due to Orai1/STIM1 interaction may induce canonical transient receptor potential channel 1 (TRPC1) translocation to the plasma membrane, where it is activated by STIM1. All these events give rise to store-operated calcium entry (SOCE). Besides the main pathway underlying SOCE, which mainly involves Orai1 and TRPC1 activation, STIM1 modulates many other plasma membrane proteins in order to potentiate the influxof Ca^2+^. Furthermore, it is now clear that STIM1 may inhibit Ca^2+^ currents mediated by L-type Ca^2+^ channels. Interestingly, STIM1 also interacts with some intracellular channels and transporters, including nuclear and lysosomal ionic proteins, thus orchestrating organellar Ca^2+^ homeostasis. STIM1 and its partners/effectors are significantly modulated in diverse acute and chronic neurodegenerative conditions. This highlights the importance of further disclosing their cellular functions as they might represent promising molecular targets for neuroprotection.

## 1. Introduction

Stromal interacting molecule 1 (STIM1), a single transmembrane-spanning domain protein mainly residing in the endoplasmic reticulum (ER), is the unique ER Ca^2+^ sensor deputed to ER Ca^2+^ refilling [1]. Historically, STIM1 was identified in 2005 by RNA interference-based screening of proteins with known signaling motifs [2]. Molecularly, STIM1 residing into the organelle senses luminal Ca^2+^ concentration by its N-terminus with a low affinity with an apparent dissociation constant (K_d_) of ~0.2–0.6 mM [3]. This low affinity is due to the high Ca^2+^ levels within the organelle allowing the function of STIM1 as a unique ER Ca^2+^ sensor. When ER Ca^2+^ diminishes, STIM1 may diffuse to the plasma membrane regions where it interacts with the channel Orai1 through its C-terminal domain, after the unfolding of the EF-hand-sterile alpha motif (SAM). This interaction triggers a localized increase in cytoplasmic Ca^2+^ that induces SERCA pump activation to refill ER of Ca^2+^. Two alternative splicing variants of STIM1 have been cloned: the long isoform named STIM1L with an extra 106 amino acids on its C-terminus, and the short 90-kDa STIM1 molecule named STIM1S. STIM1L is highly expressed in skeletal muscle as well as in several other tissues, including spleen, lungs, liver, brain and heart but not in the kidney. In human myotubes, STIM1L is expressed during myogenesis in the proximity to the PM, thus mediating a fast SOCE [4]. STIM1L displays a stronger propensity towards canonical transient receptor potential channel 1 (TRPC1) since it is more prone to open this channel [5]. Very recently, a neuronal splice variant named STIM1B has been cloned; it is selectively targeted to presynaptic ER and is useful to replenish synaptic vesicles after Ca^2+^ depletion [6]. On the other hand, STIM2 induces a weak activation of Orai1 [7] that minimally contributes to Ca^2+^ entry and prompts a limited Ca^2+^-dependent NFAT1 activation [8]. In this respect, a functional coupling between STIM2 and STIM1 has been proposed. In fact, when ER Ca^2+^ stores are maximally depleted, STIM2 targets Orai1/STIM1 complex to ER-PM junctions promoting assembly of the channel with the AKAP79/calcineurin signaling complex that promotes NFAT1 activation [8]. However, STIM1 plays the major role in clustering and activating Orai1. Therefore, STIM1 activation governs ER Ca^2+^ refilling through store-operated calcium entry (SOCE), a ubiquitous plasma membrane mechanism first described by Jim Putney in 1986 [9]. SOCE provides Ca^2+^ signals to regulate critical cell functions in many tissues after the generation of inositol 1,4,5-triphosphate (IP_3_) in response to receptor stimulation and the release of Ca^2+^ from the ER. The molecular components involved in the process remained uncharacterized for several years. A major advancement in the field came with the identification of the calcium release-activated calcium (CRAC) channel mediating the highly Ca^2+^ selective current (I_CRAC_) in mast cells and T lymphocytes [10,11,12,13,14,15,16]. Therefore, Ca^2+^ influx mediated by CRAC channels is called SOCE, because it is regulated by the filling state of ER Ca^2+^ stores. I**_CRAC_** is a non-voltage activated, inwardly rectifying current, highly selective for Ca^2+^. Accordingly, it displays a Ca^2+^:Na^+^ permeability ratio of 1000:1 [17,18]. Of note, the single-channel conductance of CRAC is small as compared with that of other plasma membrane channels, and it has been estimated in about 24 fS in Jurkat T cells [18].

Thanks to the effort of the scientific community, much is actually known about the regulatory mechanisms of SOCE (i.e., arrangement, dynamics, stoichiometry of SOCE molecular components) not only in non-excitable cells, where it is the primary Ca^2+^ influx pathway, but also in neuronal cells. However, once the role of STIM1 has been established, elucidating the other molecular components of SOCE proved to be a challenging endeavor for almost two decades. A number of proteins have been proposed to interact with STIM1, which are located both in the plasma membrane and at different intracellular levels. Some of these proteins may be considered as molecular partners of STIM1 and include Orai, TRPC channels, L-type voltage-dependent Ca^2+^ channels, plasma membrane Ca^2+^-ATPase (PMCA), exportin1, transportin1, SERCA2, SERCA3, etc. On the other hand, many modulators of SOCE, including the myelin basic protein Golli, and some scaffold proteins helping STIM1 to interact with its partners have also been described. A complex cooperation among these molecular components occurs in different parts of the cell to orchestrate intracellular Ca^2+^ homeostasis. Of note, this equilibrated cooperation may be altered during neurodegenerative disorders, thus causing neuronal death.

Molecularly, after store depletion, STIM1 interacts with the above-mentioned proteins, directly or indirectly through specific scaffold proteins. For instance, the SOCE-mediated cytosolic Ca^2+^ increase is primarily sustained by the direct activation of Orai1 by STIM1 and then by the inhibition of PMCA activity by STIM1-TMEM20 complex (see Section 4 for more details). When ER Ca^2+^ is refilled by SERCA [19], Orai1 and STIM1 dissociate and return to their resting state. This mechanism may be altered under pathological conditions through the dysregulation of both the expression and the function of STIM1 partners, thus impairing the outcomes of their interaction.

## 2. Plasma Membrane Targets of STIM1

### 2.1. Orai and TRPC1

Depleting ER Ca^2+^ store promotes the activation of different types of Ca^2+^ currents, fostering the idea that several channels and regulatory proteins may be involved in the modulation of SOCE. However, the main mechanism underlying SOCE is the interaction between STIM1 and Orai1 associated with CRAC currents (i.e., I_CRAC_) [13]. Molecularly, the activation of Orai1 is elicited by the interaction with the STIM1–Orai-activating region (SOAR) in the C-terminal region of STIM1 [20]. There are two homologs of mammalian Orai1, namely Orai2 and Orai3, all generating SOCE [21]. Furthermore, two splice variants of Orai1 have been found: the 301 amino acids Orai1α and the short variant Orai1β, both ubiquitously expressed [22]. Interestingly, the relevance of Orai components in the process of SOCE is testified by the fact that Orai1, Orai2 and Orai3 were named after the three Horae (hours), Eunomia, Dike, Eirene, who—in Greek mythology—were the guardians of the gates of Olympus [23]. Interestingly, the Ca^2+^ selectivity of CRAC channels is not an intrinsic and immutable property of Orai1 but it is due to STIM1 gating. In fact, mutations of Orai1 may be a general mechanism for cells to tune Ca^2+^ and Na^+^ entry in a STIM1-independent manner. However, STIM1 binding modifies the structural features of the mutant channel pore, bestowing the classical Ca^2+^ selectivity of CRAC channels [24]. Another important consideration is that Orai1 may form heteropentameric complexes with Orai3 to function as arachidonate-regulated Ca^2+^ (ARC) channel, a store-independent channel that is activated by STIM1 located in the plasma membrane [25,26]. Interestingly, only the Orai1α splice variant seems to play a functional role in the ARC channels [27].

The *Drosophila* transient receptor potential (TRP) channel was the first proposed candidate of SOCE [28,29], and the TRP canonical 1 (TRPC1) was the first mammalian member of the TRPC channel family to be cloned [30,31]. TRPC1 interacts with and is activated by STIM1 by means of an electrostatic interaction thus aggregating with Orai1 and STIM1 within the same ER-PM junctions [32]. However, TRPC1 recruitment is dependent on Orai1 activity [33]. In fact, Orai1-mediated Ca^2+^ entry allows the translocation of TRPC1 into the plasma membrane where it is activated by STIM1 [34,35]. Despite the impressive data supporting the involvement of TRPC1 in SOCE, the biophysical features of the current associated with TRPC1 do not match with I**_CRAC_**. For that reason, the TRPC1-associated current was named store-operated calcium current (I**_SOC_**) to distinguish it from I**_CRAC_**. Specifically, Orai1/STIM1 mediates I**_CRAC_** that is highly selective for Ca^2+^ over Na^+^ (P**_Ca_**/P**_Na_** ~1), while TRPC1/STIM1 mediates I**_SOC_**, a cationic current with a poor Ca^2+^ selectivity. Accordingly, it shows a calcium over sodium ratio between 10 and 40 (P_Ca_/P_Na_ ~10–40) [36]. Furthermore, CRAC single-channel conductance has been estimated in about 24 fS in Jurkat T cells [18] while CPA-evoked I_SOC_ had a unitary conductance of 2.5 ± 0.3 pS in portal vein smooth muscle cells [36]. Interestingly, Orai1 is necessary for TRPC1-mediated I_SOC_, since knocking down both, or expressing dominant negative Orai1 constructs inhibits I_SOC_. Therefore, the entry of Ca^2+^ through I_CRAC_ channels recruits TRPC1 to the plasma membrane that, once activated by STIM1, may mediate I_SOC_ [37].

TRPC1 may associate with TRPC3, TRPC4 and TRPC5 subunits to form a channel with different sensitivity to STIM1 [38]. Interestingly, TRPC may also be activated independently from STIM1 by a G protein-coupled receptor [39]. In any case, the two Orai1 variants can be used as functional subunits of both CRAC and SOC channels [27].

TRPC1-mediated Ca^2+^ entry is associated with the regulation of many relevant pathophysiological functions in different tissues, including gene expression, smooth, skeletal, and cardiac muscle function, proliferation, migration, fluid secretion, mast cell degranulation, platelet aggregation and T cell activation [40,41,42,43]. Furthermore, dysregulated activity of TRPC1 may contribute to the progression of various types of cancers, including breast cancer, pancreatic cancer, glioblastoma, lung cancer, hepatic cancer, multiple myeloma and thyroid cancer [44]. Interestingly, most of these functions are associated with the so-called TRPC1 channelosome due to its interaction with other proteins [40]. TRPC1 channelosome contains other TRPC channel isoforms and also Ca^2+^ signaling proteins such as Ca_V_-1, HOMER, IP_3_R, PLC, G_αq/11_ and PMCA in order to mediate a plethora of cell functions [40].

### 2.2. Plasma Membrane Ca^2+^-ATPase (PMCA)

During T cell activation, STIM1 regulates intracellular Ca^2+^ clearance through the functional interaction with the plasma membrane Ca^2+^-ATPase (PMCA) [45,46]. In this way, STIM1 is able to govern intracellular Ca^2+^ homeostasis in T cells. Of note, elevated intracellular Ca^2+^ levels with specific spatiotemporal features play a critical role in the activation of the Ca^2+^-dependent nuclear factor of activated T cells (NFAT). In this respect, the interaction between STIM1 and PMCA regulates NFAT-dependent cytokine production via transcriptional events in Jurkat T cells [45,46]. However, the correlation between STIM1 expression and inhibition of Ca^2+^ clearance is a non-linear phenomenon, thus suggesting the involvement of additional, but yet unknown, players. Therefore, PMCA could be considered as a member of a growing list of STIM1 targets, together with Orai [21,47,48], TRPC [34,49,50] and Ca_V_1.2 [51,52] proteins. However, STIM1 modulates PMCA function also through its partner POST, which preferentially associates with PMCA4b over PMCA4a isoforms. In this latter case, STIM1/POST potentiates, rather than inhibits, PMCA4 function [53]. Interestingly, POST prevents PMCA4 from STIM1-mediated inhibition, determining the proper coupling of PMCA4 to Orai1. This promotes Ca^2+^ entry mechanism and NFAT activation in Jurkat T cells [53].

### 2.3. Na^+^/K^+^ ATPase

A role of STIM1 in determining Ca^2+^-dependent Na^+^/K^+^ ATPase downregulation in isolated alveolar epithelial cells exposed to hypoxia has been suggested [54]. In fact, during hypoxia, calcium entry via CRAC channels triggers Na^+^/K^+^ ATPase downregulation, producing alveolar epithelial dysfunction. STIM1 knockdown prevents Na^+^/K^+^ ATPase hypofunction, thus protecting hypoxic cells from fluid reabsorption impairment. However, a direct interaction between STIM1 and Na^+^/K^+^ ATPase remains unknown.

### 2.4. L-Type Voltage-Gated Ca^2+^ Channels (VGCCs)

L-type voltage-gated Ca^2+^ channels (VGCCs) are interesting targets of STIM1, whose modulation is different from that described so far. In rat cortical neurons and vascular smooth muscle cells, the depolarization-opening of L-type VGCC is inhibited by a STIM1-dependent pathway [52]. SOAR is involved in the inhibition of L-type VGCC as observed for the conventional activation of Orai by STIM1. Furthermore, to control the activity of this channel for a longer period of time, STIM1 may also trigger its internalization. Interestingly, Orai1 co-localizes with STIM1 and L-type VGCC at ER-PM junctions after store depletion, thus suggesting the involvement of the plasma membrane component of SOCE in the modulation of L-type VGCC. However, since nonconductive Orai mutants may participate in the inhibition of L-type VGCC, the involvement of the Orai channel as a scaffold protein has been postulated. The same mechanism has been shown to occur for TRPC modulation [55]. With L-type VGCC being one of the main represented channels in excitable cells, the modification of STIM1-mediated inhibition of the channel may account for several pathogenetic mechanisms not fully clarified yet.

## 3. Intracellular Targets of STIM1

### 3.1. SERCA2A and SERCA3

Among the intracellular targets of STIM1 (Table 1), the sarco/endoplasmic reticulum Ca^2+^ ATPase (SERCA) pump is probably the most relevant. In fact, the inward current generated by STIM1/Orai interaction continuously triggers the replenishment of ER Ca^2+^ store through SERCA activity [19]. In this regard, the interaction between STIM1 and SERCA2A is mainly involved in ER Ca^2+^ refilling through Orai1 recruitment [56]. However, besides its role in SOCE, STIM1 is also involved in acidic Ca^2+^ store refilling through the interaction with SERCA3 [57]. This occurs in platelets in a SOC-independent way after thrombin or thapsigargin stimulation. Of note, STIM1-SERCA3 refilling is altered in platelets from type 2 diabetic patients, thus resulting in higher intracellular Ca^2+^ levels, platelet hyperactivity and cardiovascular defects [57].

### 3.2. TRPML1

Global intracellular Ca^2+^ homeostasis depends on a complex and very well integrated network among plasma membrane ionic proteins and intracellular Ca^2+^-storing organelles, also interacting each other through discrete membrane contact sites [58,59]. Lysosomes, the tiny acidic organelles mainly deputed to cellular catabolic activities [60] and autophagy [61], are now considered important Ca^2+^-storing compartments interacting with ER and mitochondria to maintain intracellular Ca^2+^ homeostasis under both physiological and pathological conditions [62,63]. Lysosomal Ca^2+^ could be released by intracellular cues, including nicotinic acid adenine dinucleotide phosphate (NAADP), acting on two-pore channels [64], and the lysosome-enriched phosphoinositide PI(3,5)P_2_, stimulating TRPML1 [65]. The local Ca^2+^ release from lysosomesis involved in the maintenance of global Ca^2+^ signaling through the interaction with the ER [66,67]. Therefore, it is plausible that a local machinery may connect the two organelles to allow the exchange of Ca^2+^ ions. In this respect, the ER Ca^2+^ sensor STIM1 has been identified as a fine regulator of TRPML1 activity [68,69]. Thus, the interaction between STIM1 and TRPML1 allows the regulation of both lysosomal and ER Ca^2+^ homeostasis. This may occur through a bidirectional interplay between the two organelles, both able to refill with Ca^2+^ ions in both physiological and pathological conditions [69].

### 3.3. Nuclear Proteins

At the nuclear level, some proteins belonging to the nuclear transporters karyopherins have also been found to interact with STIM1 [70,71]. However, the meaning of this interaction remains unknown. In HeLa cells, the association between STIM1 and the nuclear carrier proteins exportin 1 and transportin 1 may occur under physiological conditions in a Ca^2+^-independent way. In fact, this event is observed even in the presence of EDTA [70].

However, the interaction between STIM1 and importins and exportins requires the scaffolding molecule POST (see Section 4 for more details). Interestingly, on ER Ca^2+^ depletion, POST strongly binds to the ER Ca^2+^ sensor and translocates in proximity to the nuclear envelope, thus suggesting a possible role for ER Ca^2+^ depletion in the modulation of nuclear import/export activity [71].

**Table 1 cells-10-02518-t001:** STIM1 targets at plasma membrane (in blue) and intracellular (in green) level.

STIM1 Target	Localization	Effect of the Interaction	References	
Orai	Plasma membrane	ER Ca^2+^ refilling through SOCE	[9,21,47,48]	**PLASMA MEMBRANE** **TARGETS**
TRPC1	Plasma membrane	ER Ca^2+^ refilling through SOCE in an Orai1-dependent way	[28,29,32,33,34,35,49,50,57]	
PMCA	Plasma membrane	Regulation of intracellular Ca^2+^ homeostasis in T cells; regulation of NFAT-dependent cytokine production in Jurkat T cells	[45,46,53]	
Na^+^/K^+^ ATPase	Plasma membrane	Na^+^/K^+^ ATPase downregulation during hypoxia in alveolar epithelial cells	[54]	
L-type VGCC	Plasma membrane	Inhibition of L-type VGCC	[51,52]	
SERCA2A	Endoplasmic reticulum	ER Ca^2+^ refilling through Orai1 recruitment	[56]	**INTRACELLULAR** **TARGETS**
SERCA3	Endoplasmic reticulum	Acidic Ca^2+^ store refilling in platelets in a SOC-independent way	[57]	
TRPML1	Lysosome	Regulation of lysosomal and ER Ca^2+^ homeostasis	[68,69]	
Exportin 1 and transportin 1	Nuclear envelope	?	[70]	
Importins and exportins	Nuclear envelope	Possible role in the modulation of nuclear import/export, through the scaffold protein TMEM20/POST	[71]	

## 4. STIM1 Scaffold Proteins

STIM1 can modulate several ionic partners directly or through scaffold proteins enabling the Ca^2+^ sensor to interact with multiple transporters (Table 2). In this respect, the 10-transmembrane-spanning segment protein TMEM20 located at ER level has been considered the most important STIM1 protein mediator [71]. In fact, it has also been called the partner of stromal interaction molecule 1 (POST), since it is relevantly involved in the binding of STIM1 to plasma membrane proteins including Orai1, PMCA, SERCAs, Na^+^/K^+^ ATPase, as well as nuclear proteins located at the nuclear envelope such as importins-β and exportins. At the plasma membrane level, POST-Orai1 binding is store depletion-independent. POST downregulation or overexpression does not substantially affect Orai1-mediated CRAC, thus suggesting the possibility that POST could modulate Orai1 activity in response to other physiological stimuli, independently from store depletion events. Interestingly, POST knockdown determines an increase in PMCA activity in store-depleted cells in which the STIM1-POST complex is bound to PMCA [71]. This suggests that the interaction between STIM1-POST and PMCA reduces the pump Ca^2+^ extrusion, thus determining a sustained cytosolic Ca^2+^ elevation.

For instance, suppressing TMEM20/POST expression through miR-150 determines an increase in intracellular Ca^2+^ levels after TCR stimulation in CD8^+^ T cells. Since this is essential to induce the expression of activation-associated genes, miR-150-induced reduction in TMEM20 allows naïve CD8^+^ T cells to express anergy-inducing genes, such as Casitas B lineage lymphoma b (Cbl-b), Egr2 and p27. These findings suggest the central role of TCR-mediated intracellular Ca^2+^ regulation in naïve CD8^+^ T cells [72].

Another important STIM1-interacting protein is the microtubular protein EB1, involved in the formation of STIM1-SERCA2A complex [56]. By interacting with this protein, STIM1 may recruit SERCA2A after Orai1 association thus replenishing the intracellular Ca^2+^ store by the generation of an inward current creating a local microdomain. After the complete refilling, the complex dissociates, thus silencing the inward current. Upon store depletion, STIM1 becomes strongly bound to the POST-targeted molecules SERCA, PMCA and Na^+^/K^+^ ATPase, as well as to the nuclear transporters, importins-β and exportins. Store depletion-dependent STIM1 binding to SERCA2 [56] and some karyopherins [70] have been reported previously.

## 5. Molecular Modulators of STIM1

After ER Ca^2+^ depletion, a full array of adaptor proteins allows STIM1–Orai association with optimal efficiency. Among these adaptors, CRACR2A [73], septins [74], αSNAP [75] and STIMATE [76,77] have distinct inter-dependent roles. For instance, when ER luminal Ca^2+^ decreases, STIMATE interacts with the juxtamembrane coiled-coil region termed CC1 promoting the STIM1 transition to its active conformation. Then, septins target Orai1 to ER-PM junctions where αSNAP regulates STIM-Orai stoichiometry. Among the numerous negative regulators of STIM1 (Table 2) [78,79], the ER resident protein SARAF (SOCE-associated regulatory factor) plays a fundamental role in facilitating the Ca^2+^-dependent slow inactivation of CRAC channels after the interaction with the cytoplasmic C-terminus of STIM1 [80]. Thus, SARAF contributes to protect cells from Ca^2+^ overfilling [80,81]. However, SARAF may act in a dualistic way: a brief treatment with thapsigargin significantly reduces the association between SARAF and STIM1, promoting a transient association between SARAF and Orai1 with the aim to activate the plasmalemmal channel. This contributes to shape cytosolic Ca^2+^ signals, increasing the content of intracellular Ca^2+^ stores [80,81]. Therefore, in the absence of STIM1, SARAF may exert a direct stimulation of Orai1 activity, thus restoring basal ER Ca^2+^ refilling in a situation of ER depletion [82]. Another negative regulator of STIM1 is the fragment P100 of polycystin-1, the gene product of PKD1 whose mutations result in autosomal dominant polycystic kidney disease (ADPKD). Interestingly, only the cleavage product P100 reduces SOCE via direct inhibition of STIM1 translocation [83]. Furthermore, a 58-kDa thiol oxidoreductase, ERp57, exerts a modulatory role on SOCE by regulating STIM1 oligomerization [84]. Of note, thrombocythemia or primary myelofibrosis patients carry a somatic mutation of the calreticulin gene determining defective interactions between mutant calreticulin, ERp57, and STIM1. This event may activate SOCE and generates non-physiological spontaneous cytosolic Ca^2+^ influx [85].

An interesting study identified stanniocalcin 2 (STC2), a secreted glycoprotein involved in phosphate metabolism, as a negative regulator of SOCE [86]. Of note, stanniocalcin 2 is required for thrombin-induced STIM1–Orai1 interaction and the subsequent SOCE in platelets [87]. Interestingly, platelets from STC2-deficient mice showed enhanced aggregation through the increase in Orai3, but not TRPC3 and TRPC6 [87]. STIM1 can directly interact with microtubules (MT) plus end-binding protein EB1 that plays a role in ER tubulation [88]. Besides its role as scaffold in the formation of STIM1 complex, EB1 binding is also considered a trapping mechanism restricting STIM1 to ER-PM junctions, thus preventing excessive SOCE and ER Ca^2+^ overload [89]. Finally, it has been reported that Golli-BG21, a member of the MBP (myelin basic protein) family of proteins, regulates SOCE in Tcells and oligodendrocyte precursors. Specifically, Golli interacts with the C-terminal domain of STIM1, reducing the activity of SOCCs [90].

**Table 2 cells-10-02518-t002:** Scaffold proteins and modulators of STIM1.

STIM1 Modulator	Localization	Effect	References
TMEM20/POST	ER membrane; plasma membrane (minor fraction)	Scaffold protein involved in the binding of STIM1 to plasma membrane proteins (i.e., Orai1, PMCA, SERCAs, Na^+^/K^+^ ATPase) and nuclear membrane proteins (importins and exportins)	[71]
SARAF (SOCE-associated regulatory factor)	ER membrane	It protects cells from Ca^2+^ overfilling by promoting the Ca^2+^-dependent slow inactivation of CRAC channels after the interaction with STIM1	[80,81]
Fragment P100 of polycystin-1	?	Reduction in SOCE via direct inhibition of STIM1 translocation	[83]
ERp57	ER lumen	Negative modulation of SOCE via regulation of STIM1 oligomerization	[84]
Stanniocalcin 2	Extracellular (secreted); ER lumen	Negative regulation of SOCE	[86]
EB1	Microtubules	It restricts STIM1 to ER-PM junctions, thus preventing excessive SOCE and ER Ca^2+^ overload	[89]
Golli	Cell body, nucleus and processes	By interacting with the C-terminal domain of STIM1, it negatively regulates the activity of SOCCs	[90,91]

## 6. STIM1 Partners in Neurodegenerative Diseases

The involvement of STIM1 (and its partners) in neuronal injury has been demonstrated in diverse acute and chronic degenerative conditions (Figure 1). Increased STIM1 expression precedes cell death of cortical neurons in rats exposed to lateral head rotational injury [92]. Similarly, traumatic brain injury triggers elevation of STIM1 expression that contributes to apoptotic cells death by upregulating metabotropic glutamate receptor (mGluR)1-dependent Ca^2+^ signalling [93].

Proximity ligation assay, co-immunolocalization and co-immunoprecipitation experiments performed either in situ or in vitro disclosed the existence of a physical interaction between STIM1 or STIM2 and the N-methyl-D-aspartate receptor (NMDAR)2 subunits, NR2A and NR2B [94]. The authors hypothesized that ER Ca^2+^ depletion, that typically facilitates STIM-Orai1 interaction, also reduces the formation of hetero-complexes, mainly composed of STIM1-NR2B and STIM2-NR2A, being the former instrumental to allow STIM1 translocation to the plasma membrane. Therefore, STIMs can modulate NMDAR-mediated Ca^2+^ signalling by directly interacting with receptor subunits. Alternatively, in hippocampal neurons, exposure to glutamate prompts Ca^2+^ influx through NMDARs and L-type VGCCs, resulting in the release of Ca^2+^ from the ER and activation of STIM1 [95]. Thus, NMDAR stimulation activates STIM1 to directly regulate the structural plasticity of L-type VGCC-dependent dendritic spines [95].

Thus, dysfunctions of STIM and NMDAR have been suggested to be involved in the pathogenesis of neurodegenerative diseases, although the mechanisms involved in the interplay between these two partners need to be further elucidated [96,97]. Overstimulation of NMDARs results in intracellular Ca^2+^ overload that underlies excitotoxic neuronal death in neurodegeneration occurring in brain ischemia, Huntington’s disease (HD) and Alzheimer’s disease (AD) [98,99,100,101]. In these contexts, targeting the mechanism that links NMDAR with STIM proteins might represent a promising neuroprotective strategy [95,102,103,104]. Accordingly, upregulating STIMs can protect against NMDA-induced dysfunctions of Ca^2+^ homeostasis [94]. In addition to its role in the regulation of NMDAR-mediated excitotoxic injury, STIM1 has also been shown to be involved (directly or indirectly through its partners/effectors) in detrimental mechanisms specifically activated under diverse neurodegenerative conditions (Figure 1).

Another issue that needs to be addressed when considering neurodegenerative diseases consists of evidence that Ca^2+^ channels are regulated by redox modifications of reactive cysteines typically induced through chemically (reactive oxygen species [ROS], H_2_S) or enzymatically driven electron transfer. Accordingly, SOCE is highly sensitive to redox modifications [105,106,107,108] and its function may be dramatically affected by the increased oxidative stress, typically occurring during acute (i.e., ischemic) or chronic neurodegenerative conditions. In fact, extracellular oxidants cause oxidative modifications of C195 in Orai1 and Orai2 and thereby inhibit SOCE [109], whereas redox regulation of STIM1 protein may occur through S-glutathionylation of the luminal C56 that causes Ca^2+^ store-independent clustering of STIM1 and, thus, activation of Orai channels [110]. Conversely, STIM2 oxidation has recently been shown to inhibit SOCE [111]. STIM1 function is also regulated by the ER oxidoreductase ERp57 via redox modulation of C49 and C56. The lack of these two residues caused inhibition of SOCE, whereas ERp57 knockdown caused its elevation [84]. Furthermore, studies have demonstrated that S-nitrosylation of STIM1 C49 and C56 interferes with STIM1 oligomerization and consequently leads to SOCE inhibition [112,113]. Thus, a better understanding of the crosstalk between redox modifications and STIM functions/interactions is crucial to understand the role of these calcium regulatory proteins in neurodegenerative diseases.

### 6.1. Ischemic Injury

In ischemic brain injury, reduced expression of STIM1 and Orai1 has been suggested to underlie hypoxic/ischemic neuronal death in rats undergone focal ischemia and in primary cortical neurons exposed to oxygen and glucose deprivation (OGD) followed by reoxygenation [114]. In fact, under ischemic conditions, massive ER Ca^2+^ depletion may trigger neuronal death through ER stress, while Ca^2+^ refilling through SOCE underlies neuronal protection [115,116,117]. Accordingly, ischemic tolerance induced by ischemic preconditioning (IPC) has been reported to occur via the attenuation of ER stress response in ischemic neurons, while STIM1/Orai1-dependent Ca^2+^ refilling is considered a crucial mechanism involved in IPC-induced neuroprotection in rats [114,118]. STIM1 has also been shown to interact with the lysosomal Ca^2+^ channel TRPML1 to modulate lysosomal and ER Ca^2+^ content and their interplay is strongly affected by low oxygen conditions. Elevation of the expression of both proteins during OGD followed by 8 h of reoxygenation is coincident with ROS hyperproduction in cortical neurons [69]. This may result in persistent TRPML1 Ca^2+^ release and lysosomal Ca^2+^ loss. On the other hand, TRPML1 expression and activity are significantly reduced in cortical neurons exposed to IPC thus hampering lysosomal Ca^2+^ loss [69]. Thus, another mechanism by which STIM1 affects neuronal survival/death during hypoxia consists of the modulation of lysosomal TRPML1 channel activity to regulate organellar Ca^2+^ homeostasis.

Nevertheless, depending on the specific context, SOCE can also provide excessive Ca^2+^ influx, resulting in non-excitotoxic neuronal death that has been involved in ischemic damage [119,120]. In line with this concept, numerous in vitro studies have shown that STIM2 is implicated in ischemia-induced neuronal Ca^2+^ accumulation [119,121,122] and deficiency of STIM2 confers protection against stroke in mice, likely through reduced Ca^2+^ accumulation into ER of neurons [119]. However, reduced expression of STIM1 and SARAF is associated with neuronal loss in the cortex of mice subjected to focal cerebral ischemia, whereas neuroprotection exerted by IPC prevents the reduction in SARAF (but not STIM1) induced by the insult [123]. This latter evidence could be interpreted as a compensatory mechanism to restore ER Ca^2+^ refilling in neurons in the absence of STIM1. An alternative hypothesis is based on the ability of SARAF to directly stimulate Orai1 activity, thus restoring basal ER Ca^2+^ refilling in a situation of ER depletion [82].

The “beneficial” role of STIM1 is also underscored by the evidence that in stroke-prone spontaneously hypertensive rat (SHRSP), the reduction in glial STIM1 is associated with the exaggerated sympathetic response leading to stroke. In fact, truncated STIM1 expressed in SHRSP astrocytes fails to interact with the Ca^2+^ channel TRPC1 located in the plasma membrane and impairs SOCE function [124], resulting in perturbation of downstream genes and neuronal network dysfunction. Further experimental work is needed to clarify the apparently dualistic role of STIM1 in the diverse cellular players (neurons vs. glia) of ischemic damage, also focusing on the modulatory role exerted by its partners (e.g., TRPML1, SARAF or TRPC1) to be exploited as molecular targets for ischemic stroke neuroprotection.

### 6.2. Alzheimer’s Disease (AD)

Alzheimer’s disease (AD) is the most common neurodegenerative disease, whereby the progressive neuronal demise is likely triggered by β-amyloid (Aβ) peptide accumulation [125]. Nevertheless, pharmacological approaches targeting this process have failed to block disease progression [126,127,128], suggesting that mechanisms other than Aβ peptide accumulation play a crucial pathobiological role. 

Altered Ca^2+^ homeostasis is a very early sign of cell dysfunction in AD patients [129,130,131,132] and mutant presenilins (PSEN1 and PSEN2), associated with the majority of familial AD (FAD) cases, have been implicated in AD-related Ca^2+^ dysregulation, through different mechanisms, often independent from their γ-secretase activity [133]. These include altered Ca^2+^ homeostasis in ER, through increased sensitivity of inositol trisphosphate receptors (IP_3_Rs) [134,135,136,137], increased expression and activity of ryanodine receptors (RyRs) [138], negative regulation of SERCA pump [139,140] and most notably, decreased Ca^2+^ influx via SOCE [130,140,141,142,143,144,145,146,147,148]. Accordingly, SOCE is reduced in various FAD-PSEN expressing cells [130,142], with both PSEN1 and PSEN2 involved in STIM1 downregulation [140]. PSENs modulate cellular levels of STIM proteins [149,150,151]; in fact, PSENs knockout in mouse embryonic fibroblasts results in increased STIM1 levels by affecting gene transcription or protein stability [149]. 

Alternatively, SOCE modulation has been ascribed to a γ-secretase-dependent mechanism, since (the transmembrane domain of) STIM1 represents a substrate of PSEN1-containing γ-secretase complexes [141]. Notably, decreased STIM1 levels, correlated with the progression of neurodegeneration, were observed in the brain of sporadic AD patients [152]. This is also consistent with the evidence that STIM1 expression is reduced (through a γ-secretase-independent manner) in PSEN-expressing SH-SY5Y cells and human FAD fibroblasts [140]. Disruption of Ca^2+^ homeostasis is also triggered by amyloid precursor protein (APP) gene mutations that occur in FAD. Accordingly, APP-deficient cells display elevated resting Ca^2+^ levels in the ER and exhibit delayed translocation of STIM1 to Orai1 upon depletion of Ca^2+^ store [153]. More recently, Ludewig et al. have demonstrated that STIM1 and STIM2 are upregulated in double mutants lacking both APP and its homolog APLP2, resulting in impairment of Ca^2+^ handling, of ER refill and of synaptic plasticity [154].

These results strongly suggest that aberrant STIM1 levels contribute to the pathogenesis of FAD. Interestingly, downregulation of STIM1 may compromise its role in the regulation of Ca_V_1.2, thus affecting neuronal activity in AD [51,52]. In fact, reduced STIM1 levels occurring in various AD cell models underlie defective Ca^2+^ homeostasis and learning and memory impairment in a SOCE-independent manner. This includes regulation of ER refilling through mGluR1 and 5 [155,156] and of synaptic plasticity [157], as well as Ca^2+^ overload through upregulation of L-type VGCCs and mitochondrial dysfunction [152]. A recent work has demonstrated that STIM1-deficient SH-SY5Y cells display reduced expression of the IP_3_ receptors type 3 (IP_3_R3) that is, in turn, responsible for the low mitochondrial Ca^2+^ concentration ([Ca^2+^]) and for the reduced efficiency of this organelle [158], which are common features in AD patients [159].

The resulting decrease of [Ca^2+^] in intracellular stores, mainly ER and Golgi apparatus, contributes to neuronal demise by affecting protein synthesis and folding [160,161], mitochondrial homeostasis [162,163] and/or ER-mitochondrial Ca^2+^ shuttling that maintains ATP production and prevents autophagy [162,163,164,165]. Despite having a role in neuronal function, SOCE attenuation results in the reduction in mushroom spines occurring in hippocampal neurons from FAD-PSEN1 mutant mice [150], and in increased Aβ generation in vitro [146,166,167], further highlighting its role in AD pathobiology. SOCE impairment has also been observed in astrocytes [168,169], microglia [169] and lymphoblasts from FAD patients [149], strongly suggesting a putative involvement of PSENs-related SOCE alterations in immune dysregulation occurring in AD [170].

### 6.3. Huntington’s Disease (HD)

Huntington’s disease (HD) is a progressive neurodegenerative disorder caused by a polyglutamine expansion in the huntingtin (HTT) protein, thus resulting in striatal degeneration through transcriptional dysregulation of a number of genes, including those regulating Ca^2+^ signaling as demonstrated both in experimental models and in patients [171,172,173]. At variance with AD, SOCE is elevated in HD [172,173,174,175]. In both in vitro and in vivo models, SOCE pathway is indirectly and abnormally activated by mutant HTT (mHTT) in striatal γ-aminobutyric acid (GABA)ergic medium spiny neurons (MSNs) before symptoms onset, thus highlighting the role of this Ca^2+^ current in disease progression [101,176]. In fact, synaptic loss in MSNs was ascribed to SOCE elevation due to increased expression of STIM2, while STIM1 did not appear to be involved [174]. By contrast, in human or mouse neuroblastoma cells and in primary cultures of mouse MSNs, expression of the N-terminal fragment of mHTT results in increased SOCE through STIM1 and both TRPC and Orai1 [177,178]. 

Accumulating evidence suggests crosstalk between STIMs and NMDARs that may contribute to dendritic spines pathology in HD [94,95]. Indeed, NMDAR activation triggers Ca^2+^ release from ER stores, STIM1 aggregation and SOCE stimulation in diverse experimental contexts [95,179]. Although there is evidence that compounds that stabilize abnormal SOCE may prevent dendritic spine loss in HD, further work is needed to clarify whether therapeutic strategies specifically acting on SOCE components, including STIMs, deserve further scrutiny to reverse Ca^2+^ signalling dysregulation in HD [176].

### 6.4. Parkinson’s Disease (PD)

Although the exact mechanisms involved in the pathogenesis of sporadic and familial Parkinson’s disease (PD) are still unclear, the dysregulation of ER Ca^2+^ homeostasis is believed to be crucially involved in the selective loss of dopaminergic neurons of the substantia nigra pars compacta [180,181]. Rhythmic activity of dopaminergic neurons is typically dependent on L-type Ca_V_1.3 channels, and pharmacological inhibition of these currents restores Ca^2+^-independent ‘juvenile’ pacemaking activity and protects dopaminergic neurons in animal models of the disease [182].

A recent work has demonstrated that neurotoxins (i.e., 1-methyl-4-phenylpyridinium ion, MPP**^+^**) that mimic PD lead to degeneration of dopaminergic neurons by increasing Ca^2+^ influx through Ca_V_1.3 channel, by reducing TRPC1 expression, while inhibiting stimulation-dependent STIM1-Ca_V_1.3 interaction [183]. Thus, TRPC1 suppresses Ca_V_1.3 activity by providing a STIM1-based scaffold, which is crucial for proper firing and survival of dopaminergic neurons [183,184].

In PC12 cells treated with MMP**^+^**, SOCE blockade and STIM1 depletion reduce oxidative stress, prevent mitochondrial dysfunction and increase cell viability, supporting the hypothesis that SOCC through STIM1 underlies toxicity in this model [185]. In line with these findings is the recent evidence that Sigma 1 Receptors (σ1Rs) are involved in the inhibition of TRPC1-mediated Ca^2+^ entry (by inhibiting STIM1 binding with TRPC1) in dopaminergic cells, whereby downregulation of σ1Rs expression or inhibition of σ1R activity substantially prevented MPP**^+^**-induced cell death by preserving Ca^2+^ homeostasis, mitochondrial and ER function, and inhibiting apoptosis [186]. In particular, store depletion, especially in the presence of external Ca^2+^, causes the dissociation of σ1R-STIM1 complexes while facilitating STIM1-TRPC1-Orai1 interactions, which could also play a critical role in neuronal demise [186]. Nevertheless, the aforementioned data are in contrast with the observation that exposure to MPP**^+^** decreases TRPC1 expression, its interaction with STIM1 and Ca^2+^ entry in SH-SY5Y cells [187], underscoring the need of additional experiments to clarify the role of STIM1 and its partners in PD pathogenesis.

### 6.5. Amyotrophic Lateral Sclerosis (ALS)

Amyotrophic lateral sclerosis (ALS) is a devastating and fatal neurodegenerative disease characterized by a progressive loss of both upper and lower motor neurons. Molecularly, the disease has a complex pathogenesis including the deregulation of multiple intrinsic pathways and dysfunctional Ca^2+^ homeostasis [188]. In motor neurons, a tight physical coupling between STIM1 and the main lysosomal Ca^2+^ channel TRPML1 has been demonstrated, possibly to regulate lysosomal Ca^2+^ release, autophagy defects and cell survival in β-methylamino-l-alanine (l-BMAA)-treated motor neurons [68]. However, the fate of ALS motor neurons is significantly influenced also by the neighboring glial cells. Accordingly, astrocytes derived from a superoxide dismutase (SOD)1 mutant mouse model of ALS or from brain tissue of ALS patients directly induce motor neuron loss in vitro. Interestingly, the ER calcium sensor STIM1 underlies the abnormal SOCE in SOD1^G93A^ astrocytes causing ER Ca^2+^ overload [189]. Furthermore, STIM1 accumulates in muscle fibers of ALS patients and in SOD1^G93A^ transgenic mice [190]. While STIM1 co-localizes with SERCA1 and RyR1 in normal muscle fibers, it is associated only with RyR1 in partially atrophic fibers [190]. Interestingly, this is paralleled by a prominent increase in the autophagy marker LC3-II and p62, thus showing the occurrence of a dysfunctional autophagy also in ALS muscle fibers [190].

## 7. Conclusions

SOCE elicits specific cytosolic Ca^2+^ signals that are used by both excitable and non-excitable cells for regulating critical physiological processes. The ER Ca^2+^ sensor STIM1 is responsible for SOCE machinery orchestration mainly through the interaction with Orai1 channel. However, several other STIM1 targets located in different cell compartments may serve to handle intracellular calcium concentration including plasmalemmal and intracellular channels (e.g., TRPC1, L-type voltage-dependent Ca^2+^ channels and TRPML1), and pumps (e.g., PMCA and SERCAs). Of note, the interaction between STIM1 and each of these ionic targets allows the regulation of both cytosolic and organellar Ca^2+^ homeostasis. Moreover, through the modulation of STIM1 activity and interaction, many adaptors indirectly intervene in intracellular calcium homeostasis regulation.

Several lines of experimental evidence clearly demonstrate that Ca^2+^ homeostasis and, most notably, STIM1 targets may be dysregulated in both acute and chronic neurodegenerative diseases. In fact, dysfunction of SOCE may contribute to the progression of several neurodegenerative diseases. For instance, SOCE activation appears to be neuroprotective in PD and AD, while during HD progression, neuroprotection could be achieved by reducing SOCE. With stroke injury being a multifactorial pathology dependent on the dysfunctional activity of several cell types, the role of SOCE in this neurological disease remains controversial. Therefore, dissection of the molecular components sustaining SOCE should be performed at the level of each specific cell type in order to identify the exact role played by each of them. This is true especially at the level of central nervous system (CNS) cells, where SOCE plays a dualistic role in the different forms of neurodegeneration. On the other hand, STIM1 interacts with other proteins not canonically involved in SOCE and this interaction may be altered during the neurodegenerative process. For instance, dysfunction in STIM1/TRPML1 interaction participates in ALS and stroke pathogenesis through organellar ionic dysfunction. This should be carefully taken into account since each element able to interact with STIM1 should be considered as a putative target for the development of new pharmacological entities.

Collectively, considering the relevance of Ca^2+^ dyshomeostasis in neurodegeneration, the validation of new drugs toward STIM1 targets may result in successful treatment strategies for AD, PD, HD, ALS and stroke.

## Figures and Tables

**Figure 1 cells-10-02518-f001:**
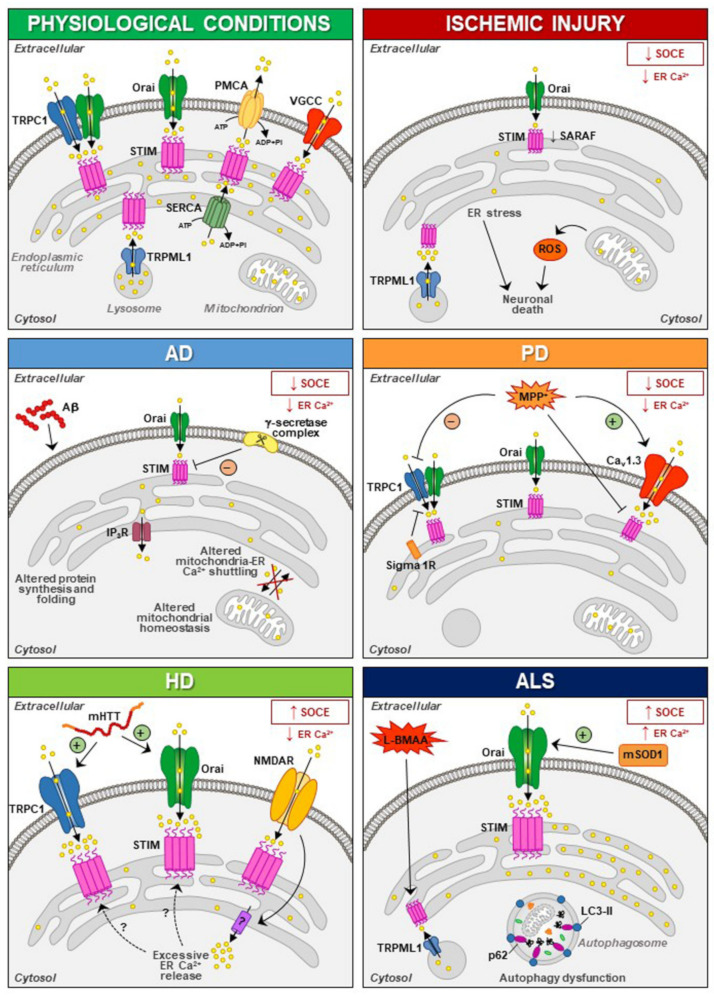
Schematic representation of STIM1 interactors at the plasma membrane and intracellular level in physiological conditions and during ischemic injury, Alzheimer’s disease (AD), Parkinson’s disease (PD), Huntington’s disease (HD) and amyotrophic lateral sclerosis (ALS). Abbreviations: ER: endoplasmic reticulum, IP_3_R: inositol trisphosphate receptor, L-BMAA: β-methylamino-L-alanine, mHTT: mutant huntingtin protein, MPP^+^: 1-methyl-4-phenylpyridinium ion, mSOD1: mutant superoxide dismutase 1, NMDAR: N-methyl-D-aspartate receptor, PMCA: plasma membrane Ca^2+^-ATPase, ROS: reactive oxygen species, SARAF: SOCE-associated regulatory factor, SERCA: sarco/endoplasmic reticulum Ca^2+^ ATPase, TRPC1: canonical transient receptor potential channel 1, TRPML1: transient receptor potential mucolipin channel 1, VGCC: voltage-gated Ca^2+^ channel.

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
