# Peer review of "Plasma Membrane and Organellar Targets of STIM1 for Intracellular Calcium Handling in Health and Neurodegenerative Diseases"

_cells, 2021, doi:10.3390/cells10102518_

Round 1

Reviewer 1 Report

Summary:

In this manuscript, Tedeschi and co-workers review two aspects of the STIM/Orai literature. The first aspect is the known protein interaction partners of STIM1 and Orai1 and the effects of these interactions on SOCE and other types of Ca2+ signaling pathways. The second is the involvements of STIM1 and Orai1 and regulators in neurodegenerative disease. The neurodegenerative discussion provides a quick and useful overview, and the associated figure does a nice job visualizing the pathways effected in the diseases. However, the protein interaction section seems somewhat incomplete and even unclear at times. Additionally, the manuscript seems to be missing a concluding section altogether that unifies each of the two main topics discussed. As such, I cannot recommend publication of the current form of the manuscript.

The specific issues that need to be addressed are listed below:

  1. Lines 18-19 of the abstract does not make sense. Please rephrase and clarify.
  2. Line 37 – The Ca2+ binding affinities reported are incorrect. The authors should quote the correct Kd values and appropriate references.
  3. Line 46 – Should state tris
  4. Lines 48-50 – The relationship between SOCE and CRAC needs to be clearly explained.
  5. Lines 87 – 100 – There are several sentences that are shown as individual paragraphs. These are too short to be paragraphs, and the information needs to be expanded to create full paragraphs or the short sentences should be combined into a single paragraph.
  6. Lines 101-102 – Do the authors mean translocation of TRPC1 into the plasma membrane rather than transport?
  7. Lines 111-112 – Do the authors mean “isoforms” instead of “variants”?
  8. Lines 106-107 – Double check the Ca2+/Na+ ratios – are these mixed up? The reference provided does not provide information on these ratios. The incorrect referencing is concerning, and all references need to be double checked.
  9. Lines 121-123 – The discussion on the ARC channels seems misplaced. It should probably be moved earlier in the manuscript, since it only involves STIM1/Orai subunits.
  10. In the authors discussion of ischemic injury it may be useful to discuss oxidative stress-induced modifications of STIM1 (i.e. S-glutathionylation) and the resulting activation.
  11. Line 360 – Should this be “SOCE”?
  12. The authors seem to be missing some STIM1 interactors (e.g. STIMATE)? Also, what about kinases (we know that STIM1 can be phosphorylated, so kinases must interact with STIM1). I would suggest a careful literature search to make the protein interaction section of the manuscript more comprehensive. We also know that STIM2 interacts with STIM1, but there is no discussion of this phenomenon in the manuscript. There may even be a database available that lists all experimentally determined protein that interact with either STIM1 or Orai1. The authors should search for this and make sure their list is complete.
  13. The involvement of STIM1/Orai1 in neurodegenerative diseases is well written; however, I think the authors need to discuss the involvement of STIM2 in mediating SOCE in neurons (see for example PMID 19843959).
  14. The manuscript needs a conclusions and future directions section that combines insights into both the protein interaction partner section and neurodegenerative disease section with unified thoughts and ideas.

Author Response

Question #1

Lines 18-19 of the abstract does not make sense. Please rephrase and clarify.

Answer:

As suggested by the Referee, the following sentence “Furthermore, Orai1–mediated Ca2+ entry allows TRPC1 transport to plasma membrane where the channel is activated by STIM1” has been rephrased and changed in: “Furthermore, Ca2+ entry due to Orai1/STIM1 interaction may induce TRPC1 translocation to the plasma membrane where it is activated by STIM1”.

Question #2

Line 37 – “The Ca2+ binding affinities reported are incorrect. The authors should quote the correct Kd values and appropriate references”.

Answer:

In accordance with this important comment, we changed the incorrect values. Ca2+ binding affinity of STIM1 EF-SAM is relatively low with an apparent dissociation constant (Kd) of almost 0.2–0.6 mM. Of course, the appropriate reference has been reported (Stathopulos P.B. et al. Stored Ca2+ depletion-induced oligomerization of stromal interaction molecule 1 (STIM1) via the EF-SAM region: An initiation mechanism for capacitive Ca2+ entry. J. Biol. Chem. 2006, 281, 35855-35862).

Question #3

Lines 48-50 – The relationship between SOCE and CRAC needs to be clearly explained.

Answer:

We thank the Referee for this request. Now a new sentence has been introduced in the Introduction section to better explain the correlation between SOCE and CRAC (please see page 2, lines 65-66).

Question #4

Lines 87 – 100 – There are several sentences that are shown as individual paragraphs. These are too short to be paragraphs, and the information needs to be expanded to create full paragraphs or the short sentences should be combined into a single paragraph.

Answer:

In accordance with this request, short sentences have been combined into single paragraphs.

Question #5

Lines 101-102 – Do the authors mean translocation of TRPC1 into the plasma membrane rather than transport?

Answer:

We thank Referee for this comment. We changed the term “transport” with “translocation”.

Question #6

Lines 111-112 – Do the authors mean “isoforms” instead of “variants”?

Answer:

In this case we referred to the Orai1 splice variants, Orai1α and Orai1β, that were introduced before the indicated lines (please see lines 101-102 of the same paragraph).

Question #7

Lines 106-107 – Double check the Ca2+/Na+ ratios – are these mixed up? The reference provided does not provide information on these ratios. The incorrect referencing is concerning, and all references need to be double checked.

Answer:

We thank the Referee for this observation. In the present version of the manuscript the correct reference reporting Ca2+/Na+ ratio has been reported (Albert A.P. and Large W.A. A Ca2+-permeable non-selective cation channel activated by depletion of internal Ca2+ stores in single rabbit portal vein myocytes. J. Physiol. 2002, 538, 717-728).

Question #8

Lines 121-123 – The discussion on the ARC channels seems misplaced. It should probably be moved earlier in the manuscript, since it only involves STIM1/Orai subunits.

Answer:

In accordance with the Referee’s comment, the sentences on ARC channels have been moved earlier in the part concerning Orai1/STIM1 interaction (please see page 3, lines 109-113).

Question #9

In the authors discussion of ischemic injury it may be useful to discuss oxidative stress-induced modifications of STIM1 (i.e. S-glutathionylation) and the resulting activation.

Answer:

As suggested, we have added a paragraph (please see page 9, lines 325-340) describing the modifications of STIM/SOCE induced by oxidative stress. Since free radicals have been implicated in a number of neurodegenerative conditions (besides ischemia), we decided to include this discussion in the general part and not specifically in the paragraph dedicated to ischemia.

Question #10

  1. Line 360 – Should this be “SOCE”?

Answer:

As suggested, we have now substituted SOCC with SOCE.

Question #11

The authors seem to be missing some STIM1 interactors (e.g. STIMATE)? Also, what about kinases (we know that STIM1 can be phosphorylated, so kinases must interact with STIM1). I would suggest a careful literature search to make the protein interaction section of the manuscript more comprehensive. We also know that STIM2 interacts with STIM1, but there is no discussion of this phenomenon in the manuscript. There may even be a database available that lists all experimentally determined protein that interact with either STIM1 or Orai1. The authors should search for this and make sure their list is complete.

Answer:

We thank the Referee #1 for this request. Accordingly, a new paragraph dedicated to the adaptor molecules of STIM1 has been added (please see page 7, lines 258-263).

Concerning the interaction between STIM1 and STIM2, new sentences on this issue have been added in the Introduction section (please see page 2, lines 54-58).

Question #12

The involvement of STIM1/Orai1 in neurodegenerative diseases is well written; however, I think the authors need to discuss the involvement of STIM2 in mediating SOCE in neurons (see for example PMID 19843959).

Answer:

We agree with the importance of mentioning the role of STIM2 in hypoxic/ischemic neurodegeneration. Therefore, we have added a sentence to the paragraph (please see page 10, lines 360-363).

Question #13

The manuscript needs a conclusions and future directions section that combines insights into both the protein interaction partner section and neurodegenerative disease section with unified thoughts and ideas.

Answer:

As suggested, a new paragraph containing conclusions and future directions has been added (please see page 15, lines 529-557).

Reviewer 2 Report

Tedeschi et al., present a comprehensive review of STIM1 and its role in neuronal calcium handling in health and disease. Overall, the authors offer a superb review of this important topic. That said, I only have few minor suggestions that should increase the appeal of this work to a wider audience:

  • Although the authors mention in passing (line 340) short-isoform of STIM1, the distinction between the long- and the short-isoforms should be introduced at the outset of the review.
  • The authors juxtapose ISOC and ICRAC on lines 105 and 106; however, they do not offer a discussion of the respective biophysical properties. The manuscript would be strengthened by addition of this basic information.
  • Lines 113-117, should also include the cardiac muscle as well as few examples of various cancer types where TRPC1 has been demonstrated to play an important role in tumor progression.

Author Response

Question #1

Although the authors mention in passing (line 340) short-isoform of STIM1, the distinction between the long- and the short-isoforms should be introduced at the outset of the review.

Answer:

We thank the Referee for this request that allows us to extend the Introduction section. Accordingly, a new small paragraph on short- and long-isoform of STIM1 has been added to the Introduction section (please see page 2, lines 44-52).

Question #2

The authors juxtapose ISOC and ICRAC on lines 105 and 106; however, they do not offer a discussion of the respective biophysical properties. The manuscript would be strengthened by addition of this basic information.

Answer:

We thank the Referee #2 for this comment that has ameliorated the manuscript. In the present version, a more explicative discussion of biophysical properties of ISOC has been added together with some basic information on Orai1/TRPC1/STIM1 interaction.

Question #3

Lines 113-117, should also include the cardiac muscle as well as few examples of various cancer types where TRPC1 has been demonstrated to play an important role in tumor progression.

Answer:

As requested, the role of TRPC1 in cardiac muscle function has been considered together with its role in tumor progression (please see page 3, lines 133-138).

Reviewer 3 Report

This is a very comprehensive review describing in detail the various plasma membrane and intracellular targets of STIM1, the nuclear proteins that interact with STIM1, and the proteins shown to be negative regulars of STIM1. Furthermore, the review describes the evidence for a role of STIM1 in various brain disorders including ischemic brain injury, Alzheimer’s disease, Huntington’s disease, Parkinson’s disease, and amyotrophic lateral sclerosis. The tables and figure in the review are a great addition. They summarise the information nicely and are easy to read. However, there are issues that need to be addressed prior to publication.

  1. The whole manuscript should be reviewed very carefully for its use of the English language. Once this is improved, the review will be much easier to read and understand.
    • Here are some examples where words are mixed up/missing. However, these examples are just to give the authors an idea of what to look for and are by no means exhaustive.
      • Line 110. ‘May be also’ should be ‘may also be’
      • Line 156. ‘Participate to’ should be ‘participate in’
      • Line 157/158 – missing words that would make the sentence make sense
      • Line 165 – is ‘working’ the correct word choice?
      • Line 170 – is ‘determining’ the correct word choice?
      • Line 184 – missing ‘a’
      • Section 3.3. I think you mean to refer to section 4 not paragraph 4?

2. There are lots of very short paragraphs with only 2-3 sentences. A paragraph should have at least 3 sentences. Perhaps these shorter paragraphs could be combined with adjacent ones to improve the flow and make the text easier to follow.

3. Section 2.4. The channel name should be specifically mentioned in the first sentence.

4. There is no reference to either table or figure throughout the text. Tables and figures should be referred to throughout the text, as appropriate, prior to their appearance in the manuscript.

5. The heading for the tables should be at the top of the table, not at the bottom.

6. Figure legend - the abbreviations (AD, PD, HD, ALS) used in the figure legend should be written in full and the abbreviations placed in brackets after each term. There are also a lot of abbreviations used in the figure that should be described in the legend, for example MPPT, L-BMAA and MHTT.

7. Section 6.5. Missing references for sentences on lines 452/453 and 455/456.

8. There is no conclusion included in the review. A conclusion bringing together the key findings of the review and future research needed is required.

Author Response

Question # 1

The whole manuscript should be reviewed very carefully for its use of the English language. Once this is improved, the review will be much easier to read and understand.

Answer:

Here are some examples where words are mixed up/missing. However, these examples are just to give the authors an idea of what to look for and are by no means exhaustive.

  • Line 110. ‘May be also’ should be ‘may also be’ – we have corrected the mistake.
  • Line 156. ‘Participate to’ should be ‘participate in’ – we have corrected the mistake.
  • Line 157/158 – missing words that would make the sentence make sense – we have rephrased the sentence.
  • Line 165 – is ‘working’ the correct word choice? – We have replaced the word “working” with “activity”.
  • Line 170 – is ‘determining’ the correct word choice? – We have replaced the verb with “resulting in”.
  • Line 184 – missing ‘a’ – We have added the missing article.
  • Section 3.3. I think you mean to refer to section 4 not paragraph 4?

Answer:

We have now substituted the word “paragraph” with “section”.

We thank the Reviewer for highlighting these mistakes/imperfections in the use of English language. We have revised the entire manuscript and corrected mistakes (including those specified above).

Question# 2

There are lots of very short paragraphs with only 2-3 sentences. A paragraph should have at least 3 sentences. Perhaps these shorter paragraphs could be combined with adjacent ones to improve the flow and make the text easier to follow.

Answer:

As suggested, we have combined shorter paragraphs to improve text flow.

Question #3

Section 2.4. The channel name should be specifically mentioned in the first sentence.

Answer:

As suggested, we have mentioned channel name in the first sentence of the section.

Question #4

There is no reference to either table or figure throughout the text. Tables and figures should be referred to throughout the text, as appropriate, prior to their appearance in the manuscript.

Answer:

References to tables and figure have been added to the text where appropriate.

Question #5

The heading for the tables should be at the top of the table, not at the bottom.

Answer:

Headings have been moved to the top of the tables.

Question #6

Figure legend - the abbreviations (AD, PD, HD, ALS) used in the figure legend should be written in full and the abbreviations placed in brackets after each term. There are also a lot of abbreviations used in the figure that should be described in the legend, for example MPPT, L-BMAA and MHTT.

Answer:

Abbreviations have been defined in figure legend.

Question# 7

Section 6.5. Missing references for sentences on lines 452/453 and 455/456.

Answer:

As requested, the missing references have been added.

Question #8

There is no conclusion included in the review. A conclusion bringing together the key findings of the review and future research needed is required.

Answer:

As suggested, a new paragraph with appropriate conclusions and future directions has been added (please see page 15, lines 529-557).

Round 2

Reviewer 1 Report

The authors have adequately addressed my concerns.

Author Response

We thank the Referee for his/her helpful comments and support. 

Reviewer 3 Report

Thank you for the making the changes. The manuscript is reading better, however there are still some issues that must be addressed prior to publication.

  1. Lines 68-68 and lines 122 – 123 - sentences need to be joined.
  2. Lines 236 – 251 – delete spaces?
  3. Table 1 – formatting issue on the right side with the word ‘Plasma’?
  4. End of section 5 – still has 2 paragraphs with only 2 sentences in each. Can they be joined or else modified?
  5. Table 2 heading – Change table heading to remove the words ‘Some of’. This doesn’t sound very professional.
  6. Figure 2 – formatting issue?
  7. Conclusion
    1. Heading – Conclusion not conclusions
    2. Line 570 – Delete abbreviation – ‘store operated calcium entry’ has already been defined.
    3. Lines 584 – 585 “Being stroke injury dependent on the defective function of several cell types…”. Suggest rewording, it’s not clear what this means.
    4. Line 588 – spelling mistake, should be ‘especially’ not ‘expecially’.
    5. Line 584 – rewording needing
    6. Line 591 – participates ‘in’ not ‘to’
    7. Line 596 – I suggest replacing ‘would’ with ‘may’.

Author Response

We thank the Referee for his/her helpful comments and support. We changed the text in accordance to his/her questions.

  1. Lines 68-68 and lines 122 – 123 - sentences need to be joined.

Answer: Thank you for this question. In the new version of the manuscript, the sentence at lines 68-69 has been added to the sentences at lines 122-123.

  1. Lines 236 – 251 – delete spaces?

Answer: The text has been now controlled for the missing  spaces

  1. Table 1 – formatting issue on the right side with the word ‘Plasma’?

Answer: We prefer to leave the text within the table as in the previous version since the changes proposed may alter the editing of the table

  1. End of section 5 – still has 2 paragraphs with only 2 sentences in each. Can they be joined or else modified?

Answer: all the paragraphs have been joined as requested

  1. Table 2 heading – Change table heading to remove the words ‘Some of’. This doesn’t sound very professional.

Answer: We have removed the indicated words.

  1. Figure 2 – formatting issue?

Answer:both the tables have been reformatted, as requested

  1. Conclusion
    1. Heading – Conclusion not conclusions –

Answer: According to Referee’s request, the word has been corrected.

  1. Line 570 – Delete abbreviation – ‘store operated calcium entry’ has already been defined.

Answer: We have deleted the abbreviation.

  1. Lines 584 – 585 “Being stroke injury dependent on the defective function of several cell types…”. Suggest rewording, it’s not clear what this means.

Answer: We have replaced the sentence with: “Being stroke injury a multifactorial pathology dependent on the dysfunctional activity of several cell types”

  1. Line 588 – spelling mistake, should be ‘especially’ not ‘expecially’.

Answer: We have corrected the mistake.

  1. Line 584 – rewording needing

Answer: We have replaced the sentence with: “Being stroke injury a multifactorial pathology dependent on the dysfunctional activity of several cell types”

  1. Line 591 – participates ‘in’ not ‘to’

Answer: We have corrected the verb in accordance to the request.

  1. Line 596 – I suggest replacing ‘would’ with ‘may’

Answer: We have replaced the word.